# Regulation Mechanism of Dopamine Receptor 1 in Low Temperature Response of *Marsupenaeus japonicus*

**DOI:** 10.3390/ijms242015278

**Published:** 2023-10-17

**Authors:** Xianyun Ren, Xueqiong Bian, Huixin Shao, Shaoting Jia, Zhenxing Yu, Ping Liu, Jian Li, Jitao Li

**Affiliations:** 1State Key Laboratory of Mariculture Biobreeding and Sustainable Goods, Yellow Sea Fisheries Research Institute, Chinese Academy of Fishery Sciences, Qingdao 266071, China; renxianyun301@163.com (X.R.); bxq20123@163.com (X.B.); jiast@ysfri.ac.cn (S.J.); liuping@ysfri.ac.cn (P.L.); 2Laboratory for Marine Fisheries Science and Food Production Processes, Laoshan Laboratory, Qingdao 266237, China; 3College of Fisheries and Life Science, Shanghai Ocean University, Shanghai 201306, China

**Keywords:** *Marsupenaeus japonicus*, low temperature, RNA interference, DAD1, apoptosis

## Abstract

Dopamine receptors (DARs) are important transmembrane receptors responsible for receiving extracellular signals in the DAR-mediated signaling pathway, and are involved in a variety of physiological functions. Herein, the D1 DAR gene from *Marsupenaeus japonicus* (*MjDAD1*) was identified and characterized. The protein encoded by *MjDAD1* has the typical structure and functional domains of the G-protein coupled receptor family. *MjDAD1* expression was significantly upregulated in the gills and hepatopancreas after low temperature stress. Moreover, double-stranded RNA-mediated silencing of *MjDAD1* significantly changed the levels of protein kinases (PKA and PKC), second messengers (cyclic AMP (cAMP), cyclic cGMP, calmodulin, and diacyl glycerol), and G-protein effectors (adenylate cyclase and phospholipase C). Furthermore, *MjDAD1* silencing increased the apoptosis rate of gill and hepatopancreas cells. Thus, following binding to their specific receptors, G-protein effectors are activated by *MjDAD1*, leading to DAD1-cAMP/PKA pathway-mediated regulation of caspase-dependent mitochondrial apoptosis. We suggest that MjDAD1 is indispensable for the environmental adaptation of *M. japonicus*.

## 1. Introduction

The diverse superfamily of GPCRs, comprising seven-transmembrane (TM) proteins, transduces external signals by binding to various ligands to initiate intracellular signaling cascades [1].GPCRs have many functions, acting as signal transducing-receptors in cellular metabolism, immune support, hormone secretion, behavioral and mood regulation, and diverse sensory activities [2].When DARs bind to their ligand, information is transduced into the cell via conformational changes, thereby activating heterotrimeric G-proteins and initiating downstream signaling pathways via the recruitment and activation of effectors [3]. Among them, the dopaminergic D1 receptor (DAD1) is the target for practically all currently available clinical antipsychotic drugs. Moreover, aberrant DA signaling correlates with many psychiatric and neurological deficits. A variety of DA receptor agonists engage different downstream effectors, according to the concept of biased agonism.

Dopamine receptors’ (DARs) downstream signal regulation affects the host defense against the external environment. However, the DAR synergistic regulation mechanisms are mostly unknown. There are five subtypes of DAR (DA1−DA5), all of which are members of the G protein-coupled receptors (GPCRs) family. DARs are further divided into D1-like and D2-like receptors based on their pharmacological profiles, signaling mechanisms, and conserved structures [4]. Subtypes DA_1_ and DA_5_ comprise D1-like receptors, which activate adenylyl cyclase (AC). AC activation leads to increased intracellular cyclic adenosine monophosphate (cAMP) levels and cell metabolic regulation, such as changes to the functions of ion channel function and GPCR desensitization, culminating in the release of neurotransmitters [5]. Subtypes DA_2_–DA_4_ comprise D2-like receptors, whose coupled signal transduction pathway inhibits AC, thereby decreasing cAMP levels. In invertebrates, DARs are vital for physiology, particularly cell signaling, learning and memory, reproduction, processing visual stimuli, mood/feeding behavior regulation, and other metabolic processes [6]. However, there have been few reports on crustacean DARs.

The economically important crustacean, kuruma shrimp *Marsupenaeus japonicus* (order Decapoda, family Penaeidae), is found throughout the Indo-West Pacific [7]. *M. japonicus* is a popular seafood produced in significant quantities by fishing and aquaculture. Currently, *M. japonicus* is farmed in several European and Asian countries, among which the largest producer is China, producing 44,548 tons in 2021 [8]. Crustaceans are cold-blooded organisms and thus face significant challenges from environmental temperature fluctuations, which result in global perturbations that modulate the reaction rates of most biological processes. Under challenge, the body’s homeostasis will mount complicated responses, including behavioral, neurological, and physiological responses, acting coordinately to rebalance homeostasis and promote organismal survival. In aquatic animals, temperature affects the function of the neuroendocrine function system, which further reduces immunity and alters their metabolism [9]. Moreover, low temperature induces apoptosis [10,11]; however, the potential toxic effect of low temperature on crustacean apoptosis and hematopoiesis is unclear.

Therefore, investigations of the immunity of shrimp must be very helpful to the healthy aquaculture of crustaceans. Herein, the DAD1 gene from *M. japonicus* (*MjDAD1*) was identified, and its role in temperature regulation was explored. We further characterized its functions and possible regulatory mechanisms to gain a detailed picture of communication between the crustacean immune and neuroendocrine systems. We explored the possible regulatory mechanism of *MjDAD1* in *M. japonicus* under low temperature stress. Furthermore, *MjDAD1* was silenced to identify cold stress-related alterations in critical neuroendocrine system factors, and the mRNA levels of apoptosis-associated genes in gills and hepatopancreas were also assessed. The results partially revealed *MjDAD1*’s functions in *M. japonicus*, and provided a mechanistic explanation for crustacean low temperature resistance.

## 2. Results

### 2.1. Characterization of the MjDAD1 Sequence

The full length cDNA sequence of *MjDAD1* was cloned using RACE technology. Sequence feature analysis of the cDNA was performed using NCBI ORF finder software (https://www.ncbi.nlm.nih.gov/orffinder/, accessed on 20 October 2021). The *MjDAD1* cDNA (GenBank accession number MK287993) has a total length of 2253 bp, including an 88 bp 5′ untranslated region (UTR) region, an ORF of 1254 bp, and a 3′ UTR of 911 bp. The cDNA contains a typical tail signal, aataaa. The 3′ UTR region contains a 26 bp polyA sequence. Starting from an ATG start codon and ending with a TAA stop codon, the ORF encodes a putative protein of 417 amino acids (Figure 1). The predicted molecular weight of MjDAD1 protein is 45.60 kDa, the theoretical pI is 8.11, and the instability coefficient is 36.35, indicating that it is a stable protein.

The prediction results of the SMATR software (http://www.cbs.dtu.dk/services/TMHMM/, accessed on 22 October 2022) showed that MjDAD1 (Appendix A) contains seven transmembrane domains at positions 40–62, 69–86, 141–163, 176–198, 213–235, 255–277, and 304–326. The figure showed that MjDAD1 has the typical structural characteristics of D1 class receptors, with a longer C-terminus.

The multiple alignment of DAD1 proteins showed strong similarity between MjDAD1 and DAD1 proteins from crustaceans. MjDAD1 has the highest sequence similarity with the protein from *Penaeus monodon* (50.46%) (Figure 2). Overall, the phylogenetic analysis showed that MjDAD1 clusters with sequences from other crustaceans, then with mollusks, and finally with vertebrates (Appendix A).

### 2.2. Tissue Expression of MjDAD1

*MjDAD1* mRNA expression in eight different *M. japonicus* tissues was assessed using qRT-PCR (Appendix A). *MjDAD1* showed the highest expression in muscle (*p* < 0.05) and the lowest in hemocytes (*p* < 0.05).

### 2.3. MjDAD1 mRNA Expression under Low Temperature Stress

*MjDAD1* mRNA expression in gills (Figure 3A) and hepatopancreas (Figure 3B) increased significantly following 10 °C and 16 °C low temperature incubation, with the highest level at 48 h (*p* < 0.05). *MjDAD1* expression did not change significantly in the 22 °C and control groups (*p* > 0.05).

### 2.4. Low Temperature Stress Effects on Intracellular Pathway Factors

#### 2.4.1. Effects of Low Temperature Stress on G Protein Effectors

The G protein effectors including AC and PLC in the gills and hepatopancreas of *M. japonicus* were determined (Appendix A). Under 10 °C low temperature incubation, the AC activity in the gills (Appendix A) and hepatopancreas (Appendix A) significantly increased from 3 to 72 h, peaking at 24 h; in the 16 °C group, the AC activity in the gills was higher than that in the control group from 24 h to 72 h, while in the hepatopancreas, the AC activity was higher than that in the control group from 3 to 72h (*p* < 0.05). In the 22 °C group, the AC activity in the gills and hepatopancreas was higher than that in the control group at 72 h (*p* < 0.05).

Under low temperature stress at 10 °C, the PLC activity in the gills (Appendix A) and hepatopancreas (Appendix A) was higher than that in the control group during the whole experiment time (*p* < 0.05). In the 16 °C group, the PLC level in the gills increased during 3−24 h, reached the minimum value at 24 h, and returned to the control groups’ level at 72 h, while in the hepatopancreas, the PLC activity was higher than that in the control group at 24 h (*p* < 0.05). In the 22 °C group, the PLC level in the gills decreased at 3 h (*p* < 0.05), and returned to the control group level at 24 h, while in the hepatopancreas, the PLC activity was higher than that in the control group at 24 h (*p* < 0.05)

#### 2.4.2. Effect of Low Temperature Stress on Second Messengers

As shown in Appendix A, low temperature had a significant effect on the second messenger concentrations in gills and hepatopancreas of *M. japonicus* (*p* < 0.05). Under low temperature stress, the cAMP level in the gills (Appendix A) was higher than that in the control group at 24 h and 72 h (*p* < 0.05). In the 10 °C group, the cAMP activity (Appendix A) in the hepatopancreas increased from 3−72 h, reaching the maximum value at 24 h; in the 16 °C group, the cAMP activity in the hepatopancreas increased at 24 h and 72 h, significantly higher than the control group (*p* < 0.05); in the 22 °C group, the cAMP activity in the hepatopancreas was higher than that in the control group at 72 h (*p* < 0.05).

Under 10 °C and 16 °C low temperature stress, the overall DAG levels in the gills (Appendix A) and hepatopancreas (Appendix A) first increased and then decreased, reaching its highest point at 24 h (*p* < 0.05). Under 22 °C stress, DAG levels in the gills and hepatopancreas first decreased and then increased, reaching its highest point at 72 h (*p* < 0.05).

Similar to DAG levels, the CaM levels in the gills (Appendix A) and hepatopancreas (Appendix A) of the 10 °C and 16 °C groups increased and then decreased, reaching its highest level at 24 h (*p* < 0.05). Under low temperature stress at 22 °C, the CaM levels in the gills and hepatopancreas gradually increased, reaching its highest level at 72 h (*p* < 0.05).

#### 2.4.3. Effect of Low Temperature Stress on Protein Kinases

Under low temperature stress at 10 °C and 16 °C, the PKA levels in the gills (Appendix A) first increased and reached the highest levels at 3 h (*p* < 0.05), and then decreased to the control level at 72 h. Under 22 °C low temperature stress, the PKA levels in the gills gradually increased, reaching a peak at 72 h (*p* < 0.05). Under low temperature stress at 10 °C and 16 °C, the PKA levels in the hepatopancreas (Appendix A) first increased and reached the highest levels at 24 h (*p* < 0.05); in the 22 °C group, the PKA levels in the hepatopancreas decreased at 24 h (*p* < 0.05), then increased to the control level at 72 h.

Under 10 °C low temperature stress, the PKC levels in the gills (Appendix A) were significantly higher than in the control group during the whole experiment time (*p* < 0.05). In the 16 °C group, the PKC levels in the gills were higher than in the control group at 24 h (*p* < 0.05), then decreased to the control level at 72 h. The PKC levels in the gills of the 22 °C group showed no significant difference compared with the control during the whole experiment. Under low temperature stress of 10 °C and 16 °C, the PKC activity in the hepatopancreas (Appendix A) was higher than that in the control group at 24 h and 72 h (*p* < 0.05), while in the 22 °C group, the PKC activity of the hepatopancreas was higher than in the control group at 72 h (*p* < 0.05).

### 2.5. The Relative Expression Ratio of MjDAD1 after MjDAD1 RNAi

The function of *MjDAD1* under low temperature stress was assessed using dsRNA interference. After injecting dsRNA into *M. japonicus*, the mRNA expression level of *MjDAD1* in gill and hepatopancreas was detected using qRT-PCR. In comparison with the NS group, *MjDAD1* expression reached its lowest value at 14 h after the injection of dsRNA, decreasing by 66% and 60% in the gills and hepatopancreas, respectively (Appendix A), indicating that *MjDAD1* was successfully silenced, and the synthesized dsRNA could be used in the RNA interference experiment.

### 2.6. Effect of MjDAD1 Interference on Tissue Damage of M. japonicus under Low Temperature Stress

#### 2.6.1. Histopathology of the Gills and Hepatopancreas after MjDAD1 Interference

The gills of *M. japonicus* are composed of gill filaments and gill segments, which include epithelial cells, keratin membranes, and gill cavities. The gills, gill filaments, and gill segments of the control group were normal, and the gill segments grew regularly on both sides of the gill filaments (Figure 4A). After 12 h of low temperature stress at 10 °C, the NC group showed thinning or disappearance of the stratum corneum, narrowing of the subepidermal space, and significant epithelial disintegration (Figure 4B). After 12 h low temperature stress at 10 °C, the *MjDAD1*-silenced group showed thickening of gill fragments, destruction of the epithelial layer, residual necrotic cells appearing in the gill cavity, and partial collapse of the entire epithelial layer of gill fragments, leaving only vacuoles composed of chitin (Figure 4C).

The hepatopancreas of *M. japonicus* is composed of hepatic tubules. The lumen of normal hepatic tubules has the shape of a quadrangle star or pentagram with a clear boundary (Figure 5A). After 12 h of low temperature stress at 10 °C, the lumen of the NC group began to degenerate, with vacuoles appearing; however, the basement membrane was relatively intact (Figure 5B). After 12 h of low temperature stress at 10 °C, the *MjDAD1*-silenced group showed severe swelling of hepatic tubules, enlargement of the lumen, and complete loss of regular star-like structure. The swelling of hepatic tubules led to adjacent liver tubules being squeezed against each other, and the boundaries of the tubules were blurred (Figure 5C).

#### 2.6.2. TUNEL Detection of Apoptosis after *MjDAD1* Interference

The TUNEL results showed that, in the gills and hepatopancreas, the 28 °C treatment group had few apoptotic cells, while in the RNAi and NC groups, the number of apoptotic cells continued to increase with the decrease in temperature. The 28 °C *MjDAD1*-silenced group and the NC group had similar numbers of apoptotic cells, while under 10 °C low temperature treatment, the *MjDAD1*-silenced group had significantly more apoptotic cells than the NC treatment group (Figure 6A,B). Figure 6C represents the statistical results of gill cell apoptosis rate (apoptotic cells/total cells). Following 12 h of low temperature stress, the apoptosis rates were 1.73% in the 28 °C NC group, 2.35% in the 28 °C RNAi group, 21.59% in the 10 °C NC group, and 33.70% in 10 °C RNAi group. Figure 6D represents the statistical results of the hepatocyte apoptosis rate. Following 12 h of low temperature stress, the apoptosis rate was 1.27% in the 28 °C NC group, 1.84% in the 28 °C RNAi group, 31.60% in the 10 °C NC group, and 51.38% in the 10 °C RNAi group.

### 2.7. MjDAD1 Silencing Affects Intracellular Signaling Pathway Factors in the Gills and Hepatopancreas under Low Temperature Stress

We detected intracellular signaling pathway factors, including protein kinases, second messengers, and G protein effectors in the gills and hepatopancreas of *M. japonicus* after low temperature exposure. The results showed that under 10 °C low temperature stress, the AC level in gills (Figure 7A) and hepatopancreas (Figure 7B) in the *MjDAD1*-silenced group increased from 3–12 h, peaking at 12 h, while being significantly lower than that in the 28 °C NC group at 48 h (*p* < 0.05). The PLC level (Figure 7C,D) in the 10 °C NC group increased from 3–12h, peaking at 12h, and in the 10 °C RNAi group the PLC level was significantly lower than that in the 28 °C NC and 10 °C NC groups.

In addition, in the 10 °C *MjDAD1*-silenced group, the levels of cAMP in the gills (Figure 8A) and hepatopancreas (Figure 8B) decreased at 3 h after low temperature exposure, and recovered to the starting concentration at 48 h. After *MjDAD1* knockdown, the DAG content in the gills (Figure 8C) and hepatopancreas (Figure 8D) of the 10 °C group was downregulated significantly from 3−12 h, but was restored to the levels in the two control groups at 48 h. In the 28 °C *MjDAD1*-silenced group and 10 °C NC group, the DAG content increased steadily from 3–12h. Similarly, the CaM content in the gills (Figure 8E) and hepatopancreas (Figure 8F) of the *MjDAD1* interference group was significantly reduced compared with that in the two control groups at 3–48 h.

Moreover, the results showed that, in the gills of the 28 °C *MjDAD1*-silenced group and the 10 °C *MjDAD1*-silenced group, the levels of PKA were gradually downregulated in the gills, and were significantly lower than those in the 28 °C NC group and 10 °C NC group at 48 h (Figure 9A). The PKA content in the hepatopancreas of the 10 °C NC group showed an initial significant increase and then decreased to the control level at 48 h (Figure 9B). When *MjDAD1* was knocked down, the PKA content in the gills (Figure 9C) of the 10 °C RNAi group gradually decreased from 12 to 48 h, which was significantly lower than that in the other groups. The PKC content in the hepatopancreas (Figure 9D) of the 10 °C *MjDAD1*-silenced group was significantly higher than that of the 28 °C NC group, and returned to the same level as that in the control groups at 48 h, while in the 28 °C *MjDAD1*-silenced group and 10 °C NC group, the PKC level was higher than that in the 28 °C NC group at 12 h and 48 h (*p* < 0.05).

### 2.8. MjDAD1 Silencing Affects Apoptosis-related Gene Expression in M. japonicus under Low Temperature Stress

RNA interference showed that, when *MjDAD1* was downregulated, the expression levels of *Bcl2* expression in the gills (Figure 10C) and hepatopancreas (Figure 10D) were also downregulated in the 10 °C low temperature group, while the expression levels of *p53* (Figure 10A,B) and *cas3* (Figure 10E,F) were significantly upregulated after *MjDAD1* silencing (*p* < 0.05). We speculated that *MjDAD1* had a negative regulatory effect on *Bcl2* and a positive feedback regulation effect on *p53* and *casp3*.

## 3. Discussion

We isolated and characterized the full length cDNA sequence of *MjDAD1*. The putative *Mj*DAD1 protein has high homology (50.46%) with DAD1 from *Penaeus monodon*. *Mj*DAD1 contains seven transmembrane domains, corresponding to G protein coupled receptors, and *Mj*DAD1 has high similarity with other marine invertebrates in the transmembrane domain, indicating that *Mj*DAD1 has been highly conserved during the evolution process. These analyses indicated that we had successfully cloned the full length cDNA of *MjDAD1* from *M. japonicus*.

The eyestalk is an important neuroendocrine regulatory organ in crustaceans. *MjDAD1* is highly expressed in the eyestalk, indicating its involvement in neuroendocrine and immune regulation of *M. japonicus*. Under low temperature stress, *MjDAD1* mRNA levels increased markedly within 3 h, suggesting that *Mj*DAD1 might participate in the cold stress response of *M. japonicus*. The expression level of the D2 receptor (DRD4) in *Litopenaeus vannamei* changed under ammonia nitrogen exposure, and the expression of dopamine receptors *EsDAD1* and *EsDAD2* in *Eriocheir sinensis* also changed with the decrease in salinity during the desalination process [12,13]. These observations indicated that dopamine receptors have important functions in crustaceans’ responses to environmental stimuli and the maintenance of internal homeostasis.

Studies have shown that DARs, following binding to their membrane receptors, modulate shrimp immune defense by regulating intracellular signaling transduction pathways (nuclear transcription factors-protein kinases), ultimately affecting proteins related to immunity [14,15,16]. D1-like receptors activate AC to positively regulate the intracellular cAMP content [17,18]. cAMP activates PKA, which in turn, phosphorylates nuclear and cytoplasmic proteins to regulate gene expression [19,20]. Our results indicated that, after low temperature stress, the contents of PKC, PKA, CaM, DAG, cAMP, PLC, and AC in the gills and hepatopancreas of *M. japonicus* showed an overall trend of increase and then decrease. Previous studies have shown that cAMP and DAG, as second messengers, affect specific immunity by regulating PKA and PKC activities [21,22]. In the blue crab *Callinectes sapidus*, D1-like dopamine receptors increased the production of cAMP in the posterior gills [23]. In the present study, *MjDAD1* silencing significantly decreased the levels of G protein effectors (PLC and AC), intracellular second messengers (DAG, CaM, and cAMP), and PKA, while the content of PKC increased significantly, which demonstrate that MjDAD1 can transduce its signal via cAMP generation and Ca^2+^ mobilization in response to cold stress. Thus, we deduced that the MjDAD1 signaling pathway exists in *M. japonicus* and MjDAD1 is an ortholog of the vertebrate receptors.

Dopaminergic signaling and cellular apoptosis are believed to involve mitochondrial dysfunction and oxidative stress induced by low temperature [24,25]. Our previous study demonstrated that in the gills and hepatopancreas, low temperature increased p53 to induce apoptosis [26]. Thus, we assessed the effects of *MjDAD1* silencing on the endogenous apoptosis pathway after low temperature treatment. Mitochondrial apoptosis-related gene expressions, specifically the anti-apoptotic gene *Bcl2*, and a gene encoding an apoptotic protein downstream of the cascade, specifically *casp3*, were determined. Yin et al. [27] found that, under low temperature, apoptosis was induced through the mitochondrial pathway in *Litopenaeus vannamei*. Herein, *MjDAD1* silencing increased *p53* and casp3 expression (encoding members of the caspase-dependent mitochondrial pathway) significantly compared with those in the control; however, they returned to control levels at 48 h. This suggested that MjDAD1 is involved in the regulation of the shrimp caspase-dependent mitochondrial apoptosis pathway. Similar to our results, Pirger et al. [28] found that, in snail salivary gland cells, programmed cell death induced by dopamine was associated with caspase-3 activation and cytochrome c release. Thus, considering the findings for the endogenous pathway of apoptosis, the results of the present study suggested that, under low temperature stress, endoplasmic reticulum stress-related apoptosis and the caspase-dependent mitochondrial pathway are modulated via the D1-like mediated AC/cAMP-PKA axis.

## 4. Materials and Methods

### 4.1. Ethical Considerations

The Guidelines for the Care and Use of Laboratory Animals in China were followed when carrying out all the experiments. The Institutional Animal Care and Use Committee (IACUC) of the Yellow Sea Fisheries Research Institute, Chinese Academy of Fishery Sciences (Qingdao, China), approved the study (approval number YSFRI-2022022).

### 4.2. Experimental Animals

A commercial aquaculture market (Qingdao, Shandong Province, China) provided healthy shrimp (*M. japonicus*: body weight = 13.04 ± 0.88 g), which were allowed to acclimate for 7 days in cement tanks (area = 8 m^2^; water depth = 20 cm, non-sandy bottom) containing filtered cycling aerated seawater (salinity = 28.6‰; temperature = 28 ± 0.5 °C; pH 8.2). The shrimp received fresh clam meat twice daily during acclimation up to 24 h before their use.

### 4.3. Application of Low Temperature Stress and Collection of Samples

The shrimp were transferred to aquariums with water temperatures of 10 ± 0.2 °C, 16 ± 0.2°C, and 22 ± 0.2°C, and then sampled at different time points (0, 3, 24 and 72 h). The water temperature was controlled using an Artificial Climate Chamber (temperature range 5–50 °C, GRTE- HXB10N, Greete Energy Saving Equipment Limited Company, Weifang, China). The control water temperature was 28.0 ± 0.5 °C. Each group comprised ten shrimps, and each condition was applied as three replicates. We collected gill and hepatopancreas tissues, which were snap frozen in liquid nitrogen and placed at −80 °C before experimentation.

### 4.4. Cloning of MjDAD1

#### 4.4.1. Extraction of Total RNA and Full Length cDNA Cloning

The TRIzol Reagent (Ambion, Foster City, CA, USA) was used to extract total RNA from shrimp muscle, eyestalk, stomach, brain, thymus, gills, hepatopancreas, and hemocytes according to the supplier’s protocol. In addition, 1% agarose gel electrophoresis was used to determine the RNA quality and the RNA concentration was determine using a Nanodrop 2000 instrument (Thermo Fisher Scientific, Waltham, MA, USA). cDNA was synthesized from high quality RNA from each tissue.

Our previously published *M. japonicus* transcript database [29] provided partial *MjDAD1* cDNA sequences. A 5′ and 3′ rapid amplification of cDNA ends (RACE) kit (Takara, Shiga, Japan) was used to amplify further *MjDAD1* cDNA sequences following the supplier’s guidelines. The RACE experiments used specific primers (Table 1) and comprised initial denaturation at 94 °C for 5 min, 35 cycles at 94 °C for 30 s, 58 °C for 30 s, and 72 °C for 1 min, with a last extension at 72 °C for 10 min. The amplicons were separated using 1% agarose gel electrophoresis, the gel purified, ligated into vector pMD18-T (Takara), and sequenced commercially (Sangon, Shanghai, China). The resultant sequences were assembled into the full length *MjDAD1* cDNA.

#### 4.4.2. Analysis of the Sequence and Phylogenetic Tree Construction

The deduced protein sequence of *Mj*DAD1 was searched using BLAST at the National Center for Biotechnology Information (NCBI) database (http://blast.ncbi.nlm.nih.gov/Blast.cgi, accessed on 3 November 2021). The SMART program (http://smart.embl-heidelberg.de/, accessed on 3 November 2021) was used to predict functional sites and domains in the putative MjDAD1 sequence. The molecular weight and isoelectric point (pI) were predicted using the ProtParam tool at the ExPASy molecular biology server (http://www.expasy.org/tools, accessed on 4 November 2020). The MjDAD1 protein sequence was multiply aligned with other known DAD1 proteins using the CLUSTALW program package in the DNAMAN 8.0 software [30]. The phylogenetic tree was constructed using the neighbor-joining (N-J) method [31] in MEGA 6.0 [32] with support from 10,000 bootstrap replicates.

#### 4.4.3. The Distribution of MjDAD1 in Tissues and Analysis of Its Expression Pattern

We extracted RNA from tissues and used a PrimeScript RT Reagent Kit with gDNA Eraser (Takara) to obtain cDNA via reverse transcription. The open reading frames (ORFs) of the genes were used to design PCR primers to detect their expression levels; the control was *ACTB* (encoding β-actin) (Table 1). The relative expression levels of *MjDAD1* were determined on an ABI 7500 Real-Time PCR Detection System (Applied Biosystems, Foster City, CA, USA) using a SYBR Premix Ex Taq II kit (Takara). The reactions comprised 5 μL of SYBR Premix Ex Taq II (2×), 2 μL of cDNA, 0.4 μL of specific primers (10 mol/L), 0.2 μL of ROX Reference Dye II (50×), and 3.4 μL of RNase-free water. The cycling comprised initial denaturation at 95 °C for 30 s; 40 cycles of 95 °C for 5 s, 60 °C for 34 s, and 95 °C for 15 s; then 1 min at 60 °C and 15 s at 95 °C. Each sample was analyzed repeatedly to ensure that the data were reliable. The 2^−∆∆Ct^ method [33] was employed to determine the relative expression of *MjDAD1*.

### 4.5. Short Interfering RNA (siRNA) Analysis

This study conducted temperature experiments at 10 °C and 28 °C gradients after siRNA interference. Each temperature gradient was divided into two groups, namely the RNAi group and the negative control (NC) treatment group. Shrimp were evenly divided into groups according to their respective amounts, and siRNA was injected at a rate of 1 µg/g at their fourth tail segment. The control group was injected with an NC negative control (meaningless double-stranded RNA), and gill and hepatopancreas tissues were collected from nine shrimp and stored in liquid nitrogen for RNA extraction at 3, 12, and 48 h (which were injected again at 24 h).

### 4.6. Histological Examination and TUNEL Assay

Gill and hepatopancreas samples for histological examination were prepared according to previously published methods [34,35]. In brief, the gill and hepatopancreas samples were stored for 24 h in 10% neutral formalin, paraffin-embedded, sectioned at 5 μm, and subjected to hematoxylin and eosin staining before being examined under an optical microscope (Eclipse 80i/90i, Nikon, Tokyo, Japan).

Gill tissues were subjected to terminal deoxynulceotidyl transferase nick-end-labeling (TUNEL) staining employing an In Situ Cell Death Detection Kit (Roche, Basel, Switzerland) following the supplier’s guidelines. In summary, the cell sections were deparaffinized, rehydrated, and digested with proteinase K for 30 min. The sections were added to the TUNEL reaction mixture and incubated in a humidified chamber at 37 °C for 1 h. Subsequently, the sections were rinsed using phosphate-buffered saline (PBS), stained using 3,3′-diaminobenzidine (DAB), subjected to Mayer-hematoxylin counterstaining, viewed under a microscope, and photographed. Image-Pro Plus 6.0 (Media Cybernetics, Rockville, MD, USA) was used to quantitatively analyze the staining intensity.

### 4.7. Protein Kinases and Intracellular Second Messengers Determination

#### 4.7.1. Supernatant Preparations

Gill and hepatopancreas samples (0.1 g) were homogenized in buffer at 0 °C, composed of 20 mM Tris-HCl (pH 7.6), 10% (*v*:*v*) glycerol, 1.0 mM dithiothreitol, and 1.5 mM EDTA. The debris was removed by centrifugation at 12,000× *g* (4 °C, 5 min). The supernatants were further centrifuged at 3000× *g* (4 °C, 25 min) and the obtained supernatants were analyzed for their concentrations of PKC, PKA, CaM, DAG, cAMP, PLC, and AC, and their protein contents.

#### 4.7.2. Assay for Intracellular Signaling Transduction Factors

The enzyme-linked immunosorbent assay method was used to determine the intracellular levels of protein kinases and second messengers. The AC, PLC, cAMP, CaM, DAG, PKA, and PKC concentrations were determined using shrimp AC (BP-E94058), shrimp PLC (BP-E94030), shrimp cAMP (BPE94058), shrimp CaM (BPE94030), shrimp DAG (BPE94095), shrimp PKA (BPE94011), and shrimp PKC (BPE94012) ELISA kits (Shanghai Lengton Bioscience Co., LTD, Shanghai, China). Their concentrations were determined following the supplier’s guidelines.

### 4.8. Quantitative Real-Time Reverse Transcription PCR (qRT-PCR)

Trizol was used to extract RNA from the samples following the supplier’s protocol (Roche, San Francisco, CA, USA). HiScript II Q RT SuperMix for qPCR (+gDNA wiper) kit (Vazyme, Jiangsu, China) was used to produce single-stranded cDNAs from the RNA, which were used as templates for the quantitative real-time PCR (qPCR) step of the qRT-PCR protocol. An Applied Biosystems™ 7500 Real-Time PCR instrument (ABI, Foster City, CA, USA) with ChamQ SYBR qPCR Master Mix (High ROX Premixed) kit (Vazyme) were used to carry out the qPCR reactions. Table 1 shows the primers for *Mjp53* and *ACTB* (internal control). The reaction conditions comprised: 10 min at 95 °C; 40 cycles of 95 °C for 30 s and 60 °C 34 s; 95 °C for 5 s; 60 °C for 1 min; and 95 °C for 15 s. The 2^−∆∆Ct^ method [33] was employed to determine the relative gene expression levels.

### 4.9. Statistical Considerations

Data are shown as the mean ± standard deviation (SD) of three independent experiments. Significant differences between control and experimental individuals were determined using one-way analysis of variance (ANOVA), followed by Dunnett’s test. A *p* value less than 0.05 was considered to indicate statistical significance. SPSS 23.0 for Windows (IBM Corp., Armonk, NY, USA) was employed for all statistical tests.

## 5. Conclusions

In summary, we cloned an *M. japonicus* DAR gene, *MjDAD1*, whose expression was upregulated markedly by low temperature stress. *MjDAD1* effects DA signaling, which alters PLC and AC levels, resulting in CaM, DAG, and cAMP concentration changes. Silencing of *MjDAD1* resulted in downregulated *Bcl2* expression in the hepatopancreas. Likewise, silencing of *MjDAD1* led to low temperature-induced activation of apoptosis signaling pathway-associated genes. Ultimately, PKA/PKC induced changes to nuclear transcription factors, which regulated apoptosis. These results implied that D1-like receptors, via their positive association with the cAMP-PKA-CREB signaling pathway, have synergistic functions in the *M. japonicus* neuroendocrine immune (NEI) system.

## Figures and Tables

**Figure 1 ijms-24-15278-f001:**
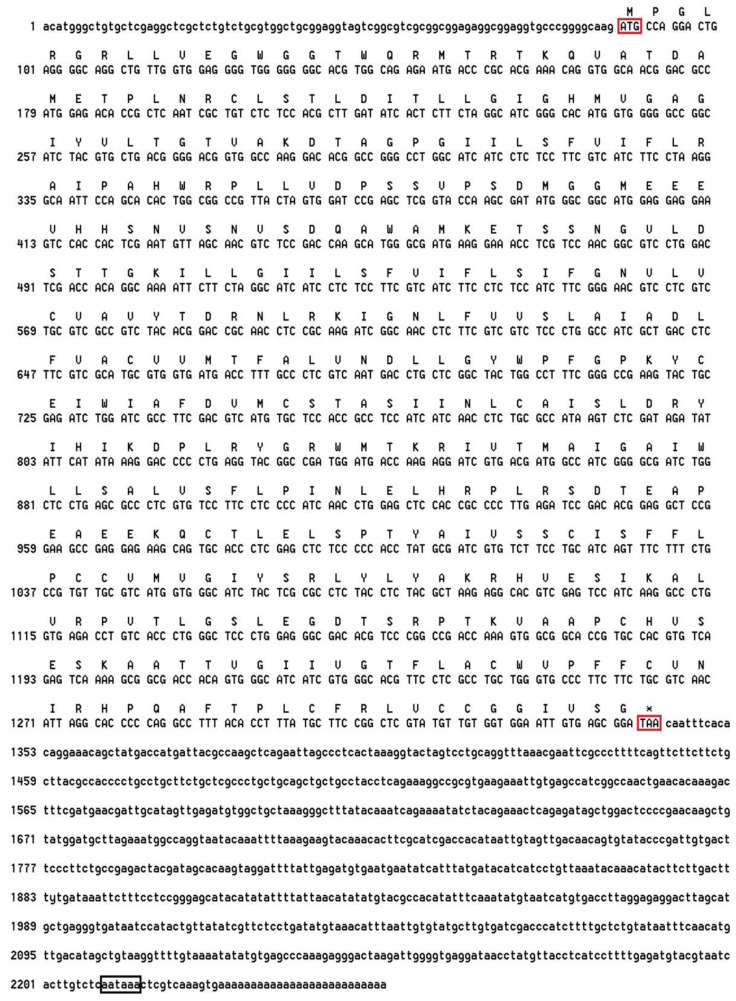
MjDAD1 nucleotide sequence and deduced amino acids sequence. The initiation codon (ATG) is marked with a red box. An asterisk (*) indicates the termination codon (TAA). The poly–A-tail signal (aataaa) is marked with a black box.

**Figure 2 ijms-24-15278-f002:**
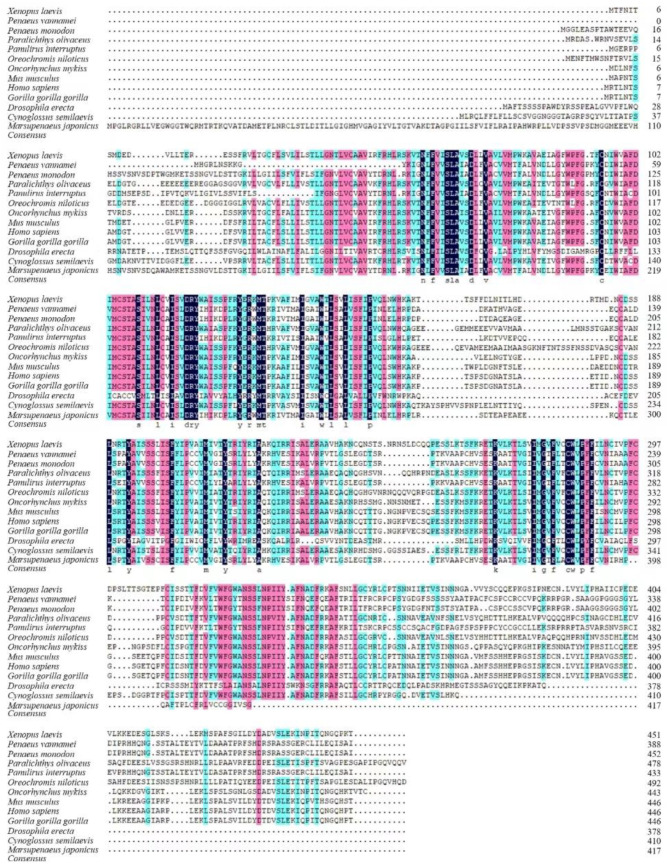
Multiple alignment of the deduced amino acid sequence of MjDAD1 with DA receptors from other species. Black boxes indicatesequence identity between amino acid sequences, pink boxes indicate the amino acid with high similarity, and blue boxes indicate the amino acid with middle similarity.

**Figure 3 ijms-24-15278-f003:**
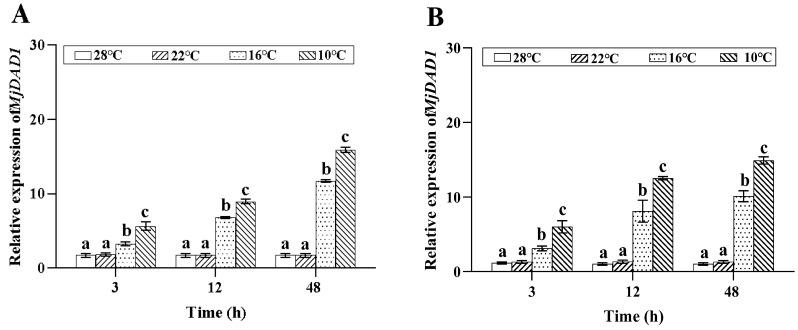
Changes in the relative expression of *MjDAD1*gene in the *M. japonicus* gills (**A**) and hepatopancreas (**B**) under low temperature stress. The data are shown as the means ± SD (n ≥ 3). Different letters on the bar chart indicate significant differences (*p* < 0.05).

**Figure 4 ijms-24-15278-f004:**
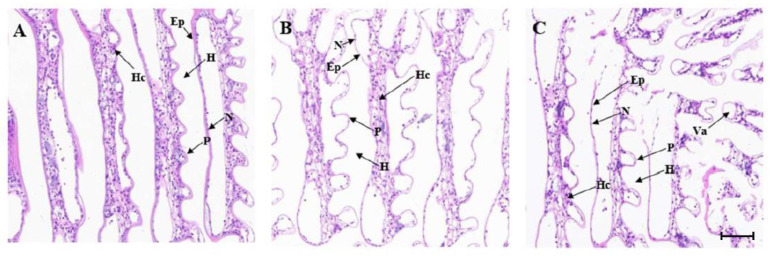
Histological structure of the gills of *M. japonicus* after *MjDAD1* interference. Note: (**A**): Control group (28 °C) (×400); (**B**): 10 °C NC stress for 12 h (×400); (**C**): 10 °C RNAi stress for 12 h (×400). Scale bar = 100 μm. (Ep) Epithelial Cell, (H) Hemocoel, (Hc) Hemocyte, (N) Epithelial nucleus, (P) Pillar cell, (Va) Vacuole.

**Figure 5 ijms-24-15278-f005:**
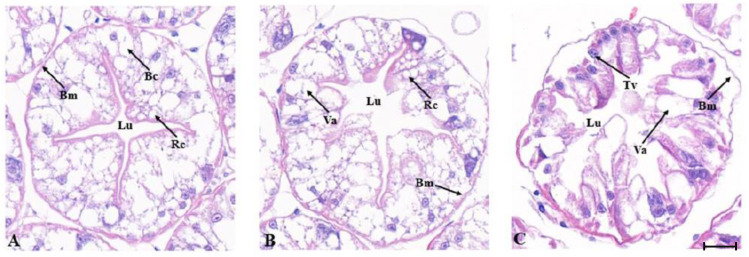
Histological structure of the hepatopancreas of *M. japonicus* after *MjDAD1* interference. Note: (**A**): Control group (28 °C) (×400); (**B**): 10 °C NC stress for 12 h (×400); (**C**): 10 °C RNAi stress for 12 h (×400). Scale bar = 100 μm. (Bc) B cells, (Bm) Basement membrane, (Lu) Lumen, (Rc) R cells, (Tv) Transferred vacuole, (Va) Vacuole.

**Figure 6 ijms-24-15278-f006:**
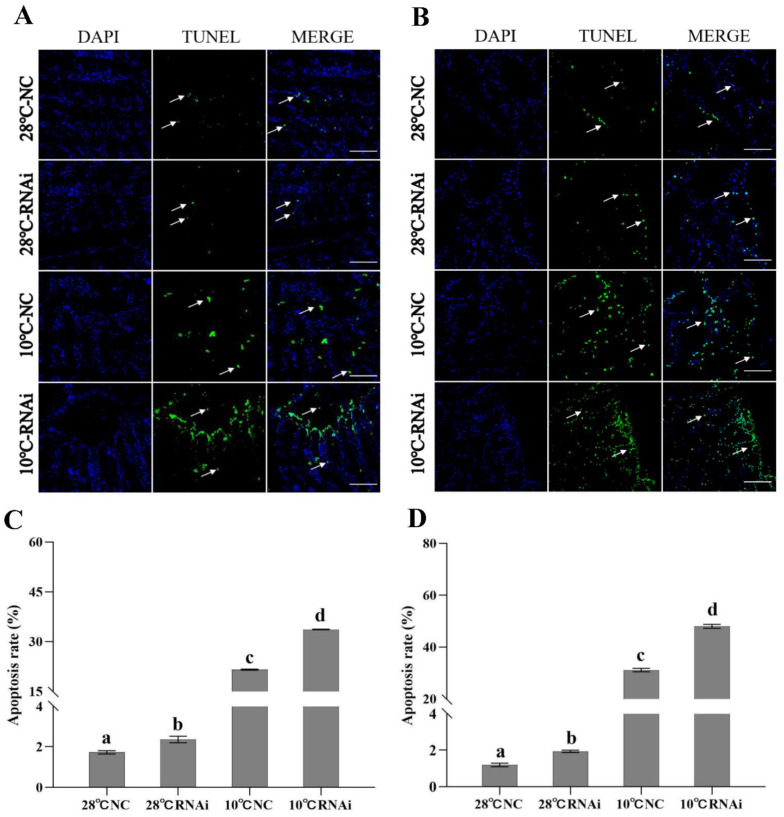
Apoptosis detection after *MjDAD1*interference. (**A**): Detection of apoptosis in the gill tissue; (**B**): Detection of apoptosis in hepatopancreatic cells; (**C**): Statistical analysis of apoptosis in gill tissue; (**D**): Statistical analysis of hepatopancreatic cell apoptosis. Apoptotic cells were detected by TUNEL staining (green) and the nuclei by DAPI staining (blue). Scale bar = 20 μm. Arrows point towards apoptotic cells. The data are shown as the means ± SD (n ≥ 3). Different letters on the bar chart indicate significant differences (*p* < 0.05).

**Figure 7 ijms-24-15278-f007:**
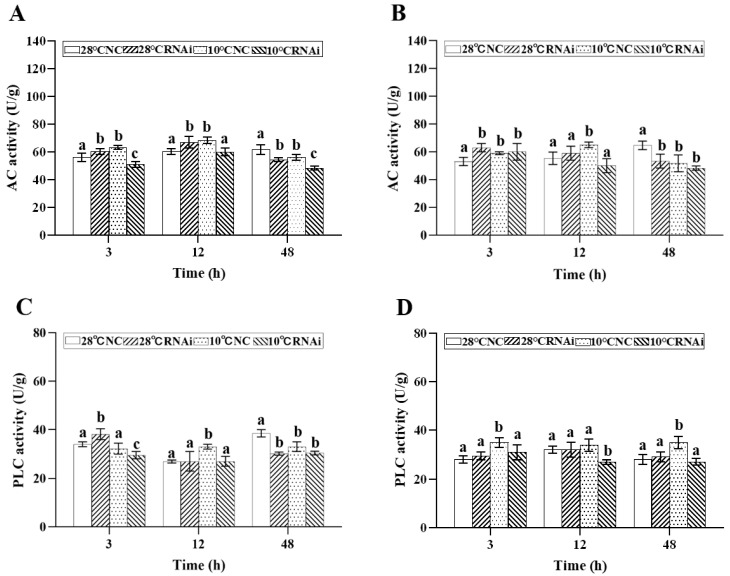
Changes in AC and PLC levels in the gills (**A**,**C**) and hepatopancreas (**B**,**D**) after RNA interference. The data are shown as the means ± SD (n ≥ 3). Different letters on the bar chart indicate significant differences (*p* < 0.05).

**Figure 8 ijms-24-15278-f008:**
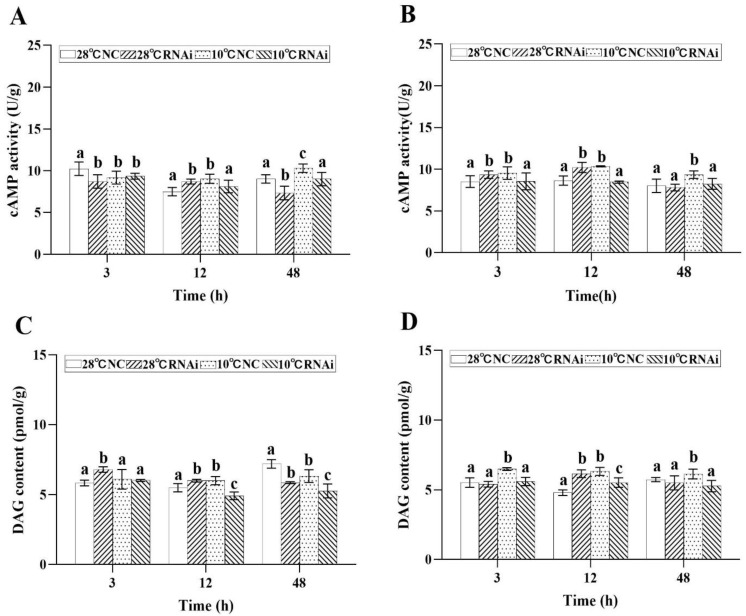
Changes in cAMP, DAG, and CaM levels in the gills (**A**,**C**,**E**) and hepatopancreas (**B**,**D**,**F**) after RNA interference. The data are shown as the means ± SD (n ≥ 3). Different letters on the bar chart indicate significant differences (*p* < 0.05).

**Figure 9 ijms-24-15278-f009:**
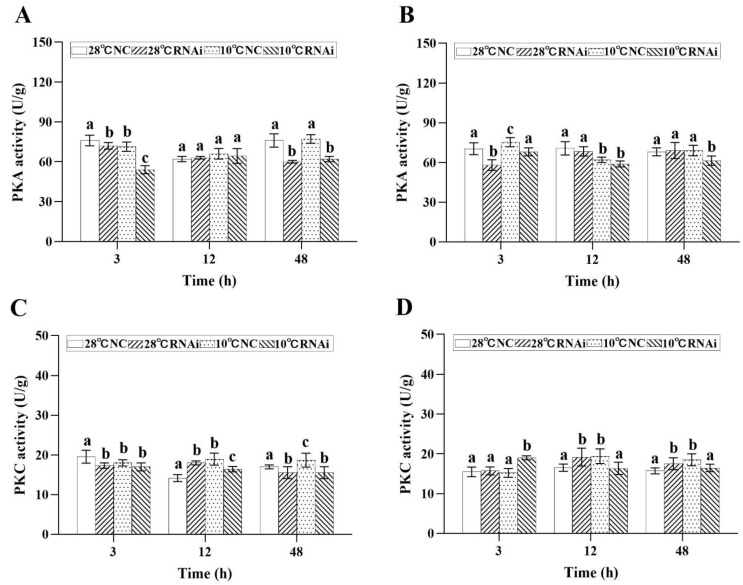
Changes in PKA and PKC levels in the gills (**A**,**C**) and hepatopancreas (**B**,**D**) after RNA interference. The data are shown as the means ± SD (n ≥ 3). Different letters on the bar chart indicate significant differences (*p* < 0.05).

**Figure 10 ijms-24-15278-f010:**
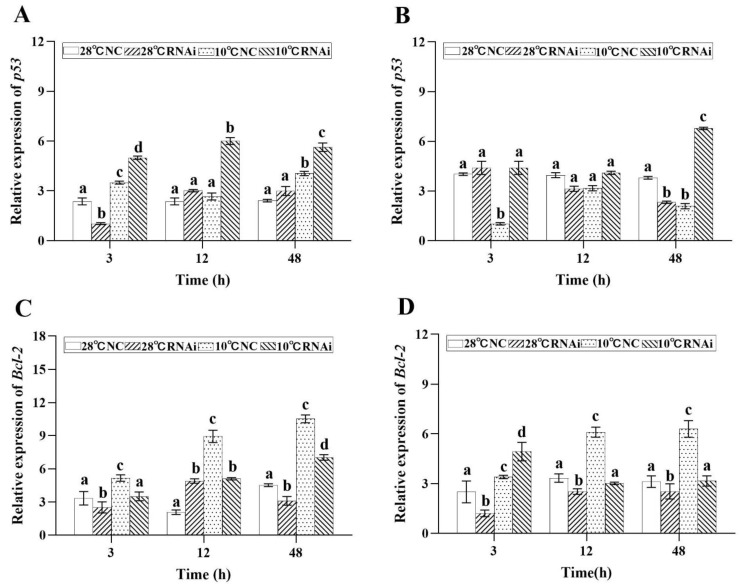
Relative expression change of *p53*, *Bcl2*, and *casp3* genes in the gill (**A**,**C**,**E**) and hepatopancreas (**B**,**D**,**F**) of *M. japonicus* after *MjDAD1* interference. The data are shown as the means ± SD (n ≥ 3). Different letters on the bar chart indicate significant differences (*p* < 0.05).

**Table 1 ijms-24-15278-t001:** Primers used in cloning and characterizing the *MjDAD1* gene.

Primer	Sequence	Usage
*MjDAD1* F1	ATTCCCGACATCGTTTTCAAGGTGC	3′ RACE
*MjDAD1* F2	CAGTTCTTCTTCTGCTTACGCCACCC	3′ RACE
*MjDAD1* R1	GGCAAAGGTCATCACCACGCA	5′ RACE
*MjDAD1* R2	AGGAAGATGACGAAGGAGAGGATGATGC	5′ RACE
UPM(short)	CTAATACGACTCACTATAGGGC	RACE
UPM(long)	CTAATACGACTCACTATAGGGCAAGCAGTGGTATCAACGCAGAGT	RACE
M13 F	GTAAAACGACGGCCAGT	colony PCR
M13 R	CAGGAAACAGCTATGAC	colony PCR
NUP	AAGCAGTGGTATCAACGCAGAGT	RACE
dsDAD1 F	GCUUGAUAUCAAUCUUCUATT	RNAi
dsDAD1 R	UAGAAGAGUGAUAUCAAGCTT	RNAi
NC F	UUCUCCGAACGUGUCACGUTT	RNAi
NC R	ACGUGACACGUUCGGAGAATT	RNAi
*MjDAD1* F	CGCCTCCATCATCAACCTCT	qRT-PCR
*MjDAD1* R	GCCATCGTCACGATCCTCTT	qRT-PCR
*MjDAD2* R	AAGCAAGCACGTCGAAACTCC	qRT-PCR
*MjBcl-2* F	TCTCAAAATGGCTCCCG	qRT-PCR
*MjBcl-2* R	GTCACTGTCGCTCACACTAC	qRT-PCR
*p53* F	CCAGTGGGTGGAGTATCA	qRT-PCR
*p53* R	TTTGTGACGACCAGCCC	qRT-PCR
*caspase-3* F	GCCTCTCACGACGCCTACAT	qRT-PCR
*caspase-3* R	GTCGCTGTGGTCTCGTT	qRT-PCR
*β-actin* F	TCCACGAGACCACATACAAC	qRT-PCR
*β-actin* R	CACTTCCTGTGAACGATTGA	qRT-PCR

## Data Availability

Not applicable.

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
