# Peer review of "Regulation Mechanism of Dopamine Receptor 1 in Low Temperature Response of Marsupenaeus japonicus"

_ijms, 2023, doi:10.3390/ijms242015278_

Round 1
Reviewer 1 Report
Dear Editor,
This manuscript described identification of the gene encoding dopamine receptor D1) DAR from Marsupenaeus japonicus (MjDAD1) together with their expression profiles upon exposure to low temperature stress and by RNA-mediated silencing. While the manuscript provides some useful information about the effect of low temperature stress on DAR-mediated signaling, most of data are too premature to support the conclusion in the manuscript, in particularly without statistical significance. Additionally, some of the figures and description in the text do not match to each other or missed point of information described in the legend. Therebefore, I recommend this paper should be rejected.
Best wishes,
Comments for the manuscript
1. In INTRODUCTION
- More information is needed for dopamine,
- rationale for studying DAD1
M&M
- how many fish were used for each experiment?
2. In results, there are too many interpretations mismatched with actual data in Fig.
3. What is the rationale for examining MjDAD from gill and hepatopancreas rather than using muscle, eyestalk, and stomach with higher level of MjDAD expression?
4. Where is the poly—A-tail signal (aataa) marked with a black box in Fig.1?
5. Where is aa sequence of MjDAD1 in Fig.2?
6. inconsistency for using MjDAD1 or MjDRD1 (Fig.3)?
7. Fig 3 showed relative expression level of MjDAD1 in Gill and hepatopancreas. What is the rationale for examining the level of MjDAD1 from gill and hepatopancreas showing a lower level of expression but not from those of muscle and eyestalk (in Discussion)?
8. in Lines 255-256.. ….with the highest level at 3 h?
: Is this true in Fig.3?
9. Too many conclusions without enough data with statistical significance.
Is there any statistical significance for the data to support such conclusion in the followings ...
: L264-266 : ..(Fig. S4B) increased first and then decreased in the 10 ℃ and 16 ℃ groups, and peaked at 24 h; while in the 22 ℃ group, the AC level in the gills and hepatopancreas increased gradually, peaking at 72 h (p < 0.05)
: in INTERPRETATION for OTHER FIGURES (S4,S5...).....
: too many conclusions insisting …. increased or decreased… without statistical significance
: L 288 .. peaking at 24 h.. & Line 290..highest level at 72 h..
:is it possible to make such a conclusion with only one point data?
: L295 ... lowest value at 24 h
: in Fig.7 no data for 24h but for 48h
: Fig 8, 9 ..
10. What is the correlation between DAD1 levels and apoptosis?
:Fig. 1 showed a higher level of DAD1 at lower temperature and a higher level of apoptosis (Fig.6C). If this is the case, how could you explain a higher level of apoptosis in RNAi sample despite of a lower level of MjDAD1.
AC and cAMP level (Fig. 7 &8)?
11. mislabels in Fig 10 vs Lines 392-393
12. Conclusion in abstract is different from the one in 5. conclusion
Engkish is OK, but there many mismatched, inaccurate information in the manuscript.
Author Response
Reviewer #1: Manuscript ID: ijms-2608856
Title: Regulation mechanism of dopamine receptor 1 in low temperature response of Marsupenaeus japonicus
This manuscript described identification of the gene encoding (dopamine receptor D1) DAR from Marsupenaeus japonicus (MjDAD1) together with their expression profiles upon exposure to low temperature stress and by RNA-mediated silencing. While the manuscript provides some useful information about the effect of low temperature stress on DAR-mediated signaling, most of data are too premature to support the conclusion in the manuscript, in particularly without statistical significance. Additionally, some of the figures and description in the text do not match to each other or missed point of information described in the legend. Therebefore, I recommend this paper should be rejected.
Comments for the manuscript
- In INTRODUCTION
- More information is needed for dopamine, - rationale for studying DAD1
Response: Thanks. We have added the more information about dopamine, and the rationale for studying DAD1 in introduction part. Lines 75-85
- In results, there are too many interpretations mismatched with actual data in Fig.
Response: Thanks. We have rewritten this part.
- What is the rationale for examining MjDAD from gill and hepatopancreas rather than using muscle, eyestalk, and stomach with higher level of MjDAD expression?
Response: Thanks. We examined MjDAD from gills and hepatopancreas for two reasons: (1) Hormones or neurotransmitters accomplish their immune functions by binding to their specific receptors on the surface of immune organ, which involves the activation of extracellular signaling moleculars. Hepatopancreas and gills in crustaceans are regarded as main immune organs (Kim and Kwak, 2022; Jiang et al., 2023). (2) Lin et al. (2022) showed that melanization response mediated by dopamine was closely related to DAR-mediated signaling pathways. Hepatopancreas controlled and mediated the synthesis of antimicrobial peptides, ROS-related antioxidant enzymes, and melanization through synergistic functions between DDC and DAR in Procambarus clarkii.
References
Jiang, H., Zhang, S., Liu, X., Li, Y., Li, H., Zhang, R., Li, X., 2023. Characterization and immune functional analysis of two new type I crustins in the oriental river prawn Macrobrachium nipponense. Aquaculture 739825. https://doi.org/10.1016/ j.aquaculture.2023.739825
Kim, W. S., Kwak, I. S., 2022. EDCs trigger immune-neurotransmitter related gene expression, and cause histological damage in sensitive mud crab Macrophthalmus japonicus gills and hepatopancreas. Fish Shellfish Immun. 122, 484-494. https://doi. org/10.1016/j.fsi.2022.02.014
Lin, S., Wang, K., Yang, B., Li, B., Shen, X., Du, Z., 2022. Dopamine receptor (DAR) and dopa decarboxylase (DDC) mediate hepatopancreas antibacterial innate immune reactions in Procambarus clarkii. Int. J. Biol. Macromol. 214, 140-151. https://doi.org/10.1016/j.ijbiomac.2022.05.200
- Where is the poly—A-tail signal (aataa) marked with a black box in Fig.1?
Response: Thanks. We have completed Figure 1. Line 107
- Where is aa sequence of MjDAD1 in Fig.2?
Response: Thanks. We have added the aa sequence of MjDAD1 in Fig.2. Line 112
- inconsistency for using MjDAD1 or MjDRD1 (Fig.3)?
Response: Thanks. We have corrected it. Line 125
- Fig 3 showed relative expression level of MjDAD1 in Gill and hepatopancreas. What is the rationale for examining the level of MjDAD1 from gill and hepatopancreas showing a lower level of expression but not from those of muscle and eyestalk (in Discussion)?
Response: Thanks. Actually we examining the level of MjDAD1 from those of muscle and eyestalk, their expression trend is not as significant as gills and hepatopancreas.
- in Lines 255-256. ….with the highest level at 3 h?
Response: Thanks. We have rewritten this part. Lines 129-136
: Is this true in Fig.3?
Response: Sorry for our careless. We have remade Fig.3. Lines 125-126
- Too many conclusions without enough data with statistical significance.
Response: Thanks. We have rewritten the result part, and added the statistical significance to support the conclusions. Lines 128-186
Is there any statistical significance for the data to support such conclusion in the followings ... (Fig. S4B) increased first and then decreased in the 10 ℃ and 16 ℃ groups, and peaked at 24 h; while in the 22 ℃ group, the AC level in the gills and hepatopancreas increased gradually, peaking at 72 h (p < 0.05)
Response: We thank the reviewer for the detailed and insightful critique of our work. We have rewritten the result part.
: in INTERPRETATION for OTHER FIGURES (S4,S5...).....
: too many conclusions insisting …. increased or decreased… without statistical significance
Response: We thank the reviewer for the detailed and insightful critique of our work. To the best of our abilities, we have tried to address these queries by careful and extensive revision, including new data and further analyses. Thanks. We thank the reviewer for the detailed and insightful critique of our work.
: L288 .. peaking at 24 h.. & Line 290..highest level at 72 h.
Response: Thanks. We have rewritten this part of result. Lines 146-161
: is it possible to make such a conclusion with only one point data?
Response: Thanks We have corrected this part. Lines 146-161
: L295 ... lowest value at 24 h
Response: Thanks. We have corrected this part. Lines 180-186
: in Fig.7 no data for 24h but for 48h
Response: Thanks. We have changed “24h” to “48h”. Lines 238-245
: Fig 8, 9 ..
Response: Thanks. We have written the rusults of Fig.8 and Fig.9.
- What is the correlation between DAD1 levels and apoptosis?
Response: MjDAD1 had a negative regulatory effect on Bcl2 and a positive feedback regulation effect on p53 and casp3. Which demonstrates MjDAD1 levels are positive correlation with apoptosis.
: Fig. 1 showed a higher level of DAD1 at lower temperature and a higher level of apoptosis (Fig.6C). If this is the case, how could you explain a higher level of apoptosis in RNAi sample despite of a lower level of MjDAD1.
Response: Thanks. When MjDAD1gene are disrupted, the body's immune function decreases, which caused the higher level of apoptosis.
AC and cAMP level (Fig.7 &8)?
Response: Thanks. We have rewritten this part. Lines 238-254
- mislabels in Fig.10 vs Lines 392-393
Response: Thanks. We have corrected the mislabeled figures. Lines 278-280
- Conclusion in abstract is different from the one in 5. Conclusion
Response: Thanks. We have unified the abstract and conclusion.
Comments on the Quality of English Language: Engkish is OK, but there many mismatched, inaccurate information in the manuscript.
Response: We thank the reviewer for the detailed and insightful critique of our work. To the best of our abilities, we have tried to address these queries by careful and extensive revision, including new data and further analyses.
Reviewer 2 Report
The manuscript, which describes the regulation mechanism of dopamine receptor 1 in low temperature response to in a crustacean species, requires some modifications.
Introduction: from line 79 to 90 the authors are referring to whom? To which organisms? Are the aberrant DA signaling that correlate with many psychiatric and neurological deficits in humans? If so, I would move this paragraph to the beginning. The authors must first describe dopamine receptors in general, and then specifically in invertebrates, and finally in crustaceans.
Materials and methods: I cannot find Table 1
Results:
· Paragraph 3.3 and figure 3: there is no correspondence between what is written in the text and the figure. What are the incubation times at low temperature?
· Paragraph 3.4 Low temperature stress’ effects on intracellular pathway factors: there is no correspondence between what is written and figures S4-S7. First, the authors must explain the meaning of the letters in the figures. The statistical analysis was performed within which groups? I think within the same time, 0, 3, 24, 48 and 72 hours. On what basis do the authors state that the different factors first increase and then decrease in the 10 and 16°C groups, reaching the highest level at 24 h? The figures, as they are made, assume a meaning that does not correspond to what is described in the text.
· I have doubts about some statistical significance, such as in fig. 7A 48h, 7D 12 and 48h etc.
· The legends of the figures must give some more information, such as the number, the statistical analysis, even if they are information already given in the text, but they must be complete and legible, without having to search for information in the text.
Author Response
Reviewer #2: Manuscript ID: ijms-2608856
Title: Regulation mechanism of dopamine receptor 1 in low temperature response of Marsupenaeus japonicus
The manuscript, which describes the regulation mechanism of dopamine receptor 1 in low temperature response to in a crustacean species, requires some modifications.
Introduction: from line 79 to 90 the authors are referring to whom? To which organisms? Are the aberrant DA signaling that correlate with many psychiatric and neurological deficits in humans? If so, I would move this paragraph to the beginning. The authors must first describe dopamine receptors in general, and then specifically in invertebrates, and finally in crustaceans.
Response: Thank you for your kind advice. We have moved this paragraph to the beginning.
Materials and methods: I cannot find Table 1
Response: Thanks. We have added Table 1. Line 382
Results: Paragraph 3.3 and figure 3: there is no correspondence between what is written in the text and the figure. What are the incubation times at low temperature?
Response: We thank the reviewer to address the problem. As the reviewer suggested, we have rewritten this part of the result. Lines 120-123
Paragraph 3.4 Low temperature stress’ effects on intracellular pathway factors: there is no correspondence between what is written and figures S4-S7. First, the authors must explain the meaning of the letters in the figures. The statistical analysis was performed within which groups? I think within the same time, 0, 3, 24, 48 and 72 hours. On what basis do the authors state that the different factors first increase and then decrease in the 10 and 16°C groups, reaching the highest level at 24 h? The figures, as they are made, assume a meaning that does not correspond to what is described in the text.
Response: Thank you for your kind advice. As the reviewer suggested, we have rewritten the result part. Lines 129-186
I have doubts about some statistical significance, such as in fig. 7A 48h, 7D 12 and 48h etc.
Response: We thank the reviewer to address the problem. We have reanalyzed the data of figure 7.
The legends of the figures must give some more information, such as the number, the statistical analysis, even if they are information already given in the text, but they must be complete and legible, without having to search for information in the text.
Response: Thank you for your kind advice. As the reviewer suggested, we have rewritten the result part.
Round 2
Reviewer 1 Report
Dear Editor,
I have checked a revised manuscript by Ren et al. I could see some progress in the revised manuscript but there are still some information and arguments that need to be clarified and confirmed to be published in IJMS.
Followings are some points that need to be clarified:
I. For aa sequence of MjDAD1 deduced from nucleotide sequence (Fig 1), alignment with other DA receptors from other species (Fig.2), and its secondary structure prediction (Fig S1).
lines 285-290 “The putative MjDAD1 protein has high homology (50.46%) with DAD1 from Penaeus monodon. MjDAD1 contains seven transmembrane domains, corresponding to G protein coupled receptors, and MjDAD1 has high similarity with other marine invertebrates in the transmembrane domain, indicating that MjDAD1 has been highly conserved during the evolution process. These analyses indicated that we had successfully cloned the full length cDNA of MjDAD1.”
a) It seems like that N-terminal domain of MjDAD1 the N-terminal domain of MjDAD appears to be extraordinarily long compared to other DA receptors (Fig.2). It also showed a much shorter C-terminal domain (& even lacking NPXXY domain conserved in all other DR listed in Fig. 2 and most of GPCRs) than those of other DR.
Q1. In this regard, have you double checked the sequence of MjDAD1 e.g. the size and sequence of full-length cDNA product to confirm the sequence around both termini? Are you sure for the sequence reported, particularly for its unusually long at N-terminus and extraordinary short & lacking the NPXXY motif, one of the most conserved motive in the 7th TM domain of GPCRs?
Q2. Have you checked your secondary structure analysis of MjDAD1?
In lines 95-97, you wrote that “The prediction results of the SMATR software showed that MjDAD1 (Fig. S1) contains seven transmembrane domains at positions 40–62, 69–86, 141–163, 176–198, 213–235, 255–277, and 304–326.”.
If your prediction is correct, how you could explain the facct that “DRY aa 234-6” sequence (one of the most conserved sequence/motif in GPCRs) might be located at end of the fifth TM domain rather than at the end of the third TM domain as shown to be located in most known GPCRs? This is another reason for checking the sequence of the N-terminus.
Q3. In lines 118-121 wrote that “MjDAD1 mRNA expression in gills (Fig. 3A) and hepatopancreas (Fig. 3B) increased significantly following 10 ℃ and 16 ℃ low temperature incubation with the highest level at 48 h ( p < 0.05). MjDAD1 expression did not change significantly in the 22℃ and control group (p > 0.05).”
However, you could not see such a clear level (~ 10 fold?) of increase at lower temperature stress (10C) in supplementary (Fig S7 for 28C NC vs 10C NC showing 2 x difference at best). Then can you still advocate the argument as in the above lines?
4. Although you indicated an overall trend of increasing and decreasing as in line 311,
: For effector enzyme (e.g. AC & PLC) and their associated second messenger concentration (e.g. cAMP & DAG, respectively), some didn’t show a correlation for the level of enzyme. How many times experiments were done for confirming the results (DADR1 level & 2nd messenger) ?
5. in line 98, what do you mean “with a shorter third intracellular ring” between which TM domains? Is this consistent with your TM prediction?
6. line 108 poly –A-tail signal (aataaa)
7. line 273 2.8
Author Response
- For aa sequence of MjDAD1 deduced from nucleotide sequence (Fig 1), alignment with other DA receptors from other species (Fig.2), and its secondary structure prediction (Fig S1).
lines 285-290 “The putative MjDAD1 protein has high homology (50.46%) with DAD1 from Penaeus monodon. MjDAD1 contains seven transmembrane domains, corresponding to G protein coupled receptors, and MjDAD1 has high similarity with other marine invertebrates in the transmembrane domain, indicating that MjDAD1 has been highly conserved during the evolution process. These analyses indicated that we had successfully cloned the full length cDNA of MjDAD1.”
- a) It seems like that N-terminal domain of MjDAD1 the N-terminal domain of MjDAD appears to be extraordinarily long compared to other DA receptors (Fig.2). It also showed a much shorter C-terminal domain (& even lacking NPXXY domain conserved in all other DR listed in Fig. 2 and most of GPCRs) than those of other DR.
Q1. In this regard, have you double checked the sequence of MjDAD1 e.g. the size and sequence of full-length cDNA product to confirm the sequence around both termini? Are you sure for the sequence reported, particularly for its unusually long at N-terminus and extraordinary short & lacking the NPXXY motif, one of the most conserved motive in the 7th TM domain of GPCRs?
Response: Thanks for your astute observation. We performed agarose gel electrophoresis and sequencing of the cDNA product of MjDAD1, and subsequent experiments were performed after the results were ideal. We used SMART to predict the secondary structure of MjDAD1 again, and the results were consistent with the previous ones. After consulting the relevant literature, it was found that SLD motif in species such as Penaeus monodon and Ixodes scapularis did have the end of the third transmembrane domain. But that fact unusually long at N-terminus and extraordinary short & lacking the NPXXY motif were found in the MjDAD1 homologous gene clone of Gryllus bimaculatus and in Scylla olivaceah. We believe that Marsupenaeus japonicus is a lower crustacean, and there are great species differences.
References
SUKTHAWORN S, PANYIM S, UDOMKIT A. Molecular and functional characterization of a dopamine receptor type1 from Penaeus monodon. Aquaculture, 2013, 380–383: 99–105.
MEYER J M, EJENDAL K F K, WATTS V J, et al. Molecular and pharmacological characterization of two D1-like dopamine receptors in the Lyme disease vector, Ixodes scapularis. Insect Biochemistry and Molecular Biology, 2011, 41(8): 563–71.
HAMADA A, MIYAWAKI K, HONDA-SUMI E, et, al. Loss-of-function analyses of the fragile X-related and dopamine receptor genes by RNA interference in the cricket Gryllus bimaculatus. Developmental Dynamics, 2009, 238(8):2025–2033.
TECHA S, THONGDA W, BUNPHIMPAPHA P, et, al. Isolation and functional identification of secretin family G-protein coupled receptor from Y-organ of the mud crab, Scylla olivacea. Gene, 2023, 848, e146900.
Q2. Have you checked your secondary structure analysis of MjDAD1?
In lines 95-97, you wrote that “The prediction results of the SMATR software showed that MjDAD1 (Fig. S1) contains seven transmembrane domains at positions 40–62, 69–86, 141–163, 176–198, 213–235, 255–277, and 304–326.”.
If your prediction is correct, how you could explain the facct that “DRY aa 234-6” sequence (one of the most conserved sequence/motif in GPCRs) might be located at end of the fifth TM domain rather than at the end of the third TM domain as shown to be located in most known GPCRs? This is another reason for checking the sequence of the N-terminus.
Response: Thanks for your astute observation. After consulting the relevant literature, it was found that the SLD motif in species such as Penaeus monodon does have the end of the third transmembrane domain. MjDAD1 is better conserved, but there is species variability, e.g., the fact that “DRY aa 234-6” (sequence one of the most conserved sequence/motif in GPCRs) might be located at end of other TM domain rather than at the end of the third TM domain were found in Scylla olivacea.
References
SUKTHAWORN S, PANYIM S, UDOMKIT A. Molecular and functional characterization of a dopamine receptor type1 from Penaeus monodon. Aquaculture, 2013, 380–383, 99–105.
TECHA S, THONGDA W, BUNPHIMPAPHA P, et, al. Isolation and functional identification of secretin family G-protein coupled receptor from Y-organ of the mud crab, Scylla olivacea. Gene, 2023, 848, e146900.
Q3. In lines 118-121 wrote that “MjDAD1 mRNA expression in gills (Fig. 3A) and hepatopancreas (Fig. 3B) increased significantly following 10 ℃ and 16 ℃ low temperature incubation with the highest level at 48 h ( p < 0.05). MjDAD1 expression did not change significantly in the 22℃ and control group (p > 0.05).”
However, you could not see such a clear level (~ 10 fold?) of increase at lower temperature stress (10C) in supplementary (Fig S7 for 28C NC vs 10C NC showing 2 x difference at best). Then can you still advocate the argument as in the above lines?
Response: Thanks for your astute observation. There are two reasons: (1) The shrimp used in the low temperature stress group and RNAi group were from different batches, individual differences can lead to differences in the results; (2) The shrimps in the RNAi groups were injected PBS, whiled in the low temperature stress group were not injected, and the differences in experimental conditions were also the reason for the differences in experimental results. Although the MjDAD1 mRNA expression levels in gills and hepatopancreas of the low temperature stress group and RNAi groups were not the same, the overall trend were the same, and were all increased under 10 ℃ and 16 ℃ low temperature incubation.
- Although you indicated an overall trend of increasing and decreasing as in line 311,
: For effector enzyme (e.g. AC & PLC) and their associated second messenger concentration (e.g. cAMP & DAG, respectively), some didn’t show a correlation for the level of enzyme. How many times experiments were done for confirming the results (DADR1 level & 2nd messenger) ?
Response: Thanks. Each experimental group has three parallels, and each parallel performs three technical repetitions were done for confirming the results (DADR1 level & 2nd messenger).
- in line 98, what do you mean “with a shorter third intracellular ring” between which TM domains? Is this consistent with your TM prediction?
Response: We thank the reviewer to adress the probllem. We have deleted this sentece. Lines 99-100
- line 108 poly –A-tail signal (aataaa)
Response: Thanks. We have corrected the sentence accordding. Lines 109-110
- line 273 2.8
Response: Thanks. We have changed the title number “3.8” to “2.8”. Line 275